# Abrogation of pathogenic attributes in drug resistant *Candida auris* strains by farnesol

Vartika Srivastava[1], Aijaz Ahmad [1,2]*

1 Clinical Microbiology and Infectious Diseases, School of Pathology, Faculty of Health Sciences, University of the Witwatersrand, Johannesburg, South Africa, 2 Infection Control, Charlotte Maxeke Johannesburg Academic Hospital, National Health Laboratory Service, Johannesburg, South Africa

* Aijaz.Ahmad@wits.ac.za, Aijaz.Ahmad@nhls.ac.za

## Abstract

*Candida auris*, a decade old *Candida* species, has been identified globally as a significant nosocomial multidrug resistant (MDR) pathogen responsible for causing invasive outbreaks. Biofilms and overexpression of efflux pumps such as Major Facilitator Superfamily and ATP Binding Cassette are known to cause multidrug resistance in *Candida* species, including *C. auris*. Therefore, targeting these factors may prove an effective approach to combat MDR in *C. auris*. In this study, 25 clinical isolates of *C. auris* from different hospitals of South Africa were used. All the isolates were found capable enough to form biofilms on 96-well flat bottom microtiter plate that was further confirmed by MTT reduction assay. In addition, these strains have active drug efflux mechanism which was supported by rhodamine-6-G extracellular efflux and intracellular accumulation assays. Antifungal susceptibility profile of all the isolates against commonly used drugs was determined following CLSI recommended guidelines. We further studied the role of farnesol, an endogenous quorum sensing molecule, in modulating development of biofilms and drug efflux in *C. auris*. The MIC for planktonic cells ranged from 62.5–125 mM, and for sessile cells was 125 mM (4h biofilm) and 500 mM (12h and 24h biofilm). Furthermore, farnesol (125 mM) also suppresses adherence and biofilm formation by *C. auris*. Farnesol inhibited biofilm formation, blocked efflux pumps and downregulated biofilm- and efflux pump- associated genes. Modulation of *C. auris* biofilm formation and efflux pump activity by farnesol represent a promising approach for controlling life threatening infections caused by this pathogen.

## Introduction

*Candida auris* has now well evolved MDR pathogen, which has caused serious outbreaks in several continents. It was first isolated from external ear of a Japanese patient in 2009 [1] and within a decade infection caused by *C. auris* has spread rapidly across six continents [1,2]. Centers for Disease Control and Prevention (CDC) has declared *C. auris* as a global threat with a report of causing several outbreaks in different countries, including United States (https://www.cdc.gov/fungal/candida-auris/tracking-c-auris.html). *C. auris* is causing serious bloodstream infections and other infections ranging from meningitis, bone infections, surgical wound infections and urinary tract infections have been reported in hospitals [3]. *C. auris*

**Data Availability Statement:** All relevant data are within the paper.

**Funding:** AA acknowledges financial support from National Research Foundation Research Development Grant for Y-Rated Researchers

(RDYR180418322304; Grant No: 116339). VS is thankful to Health Sciences Faculty Research Committee, University of the Witwatersrand for the financial support (FRC, Grant no: 0012548464101 5121105000000000000000005254).

**Competing interests:** The authors have declared that no competing interests exist.

infections are stubborn because it is resilient to most of the available antifungal drugs [4]. In one of the reports, CDC has analyzed antifungal susceptibility profile of different *C. auris* isolates and it was reported that almost all the isolates were resistant to fluconazole (FLZ) and one-third of isolates remain unaffected to amphotericin B (AmB). In 2016, Infectious Diseases Society of America (IDSA) Clinical Practice Guidelines for candidiasis has recommended Antifungal Susceptibility Testing (AFST) for all clinically relevant *Candida* isolates. Furthermore, it was suggested that any *Candida* isolate with antifungal resistance $\geq 1$ and with uncertain identity should be further tested for confirmation of *C. auris* [5]. In a study, out of 54 *C. auris* isolates from five countries, FLZ resistance was reported in 93% isolates, AmB resistance in 35% isolates, and echinocandins resistance in 7% *C. auris* isolates (around 41% *C. auris* isolates were found resistant to $\geq 2$ antimycotic class of drugs) [6]. Similarly, a recent report clears scenario of *C. auris* resistance in the U.S., 86%, 43% and 3% of first 35 patients were resistant to FLZ, AmB and echinocandins respectively [7].

Whereas, echinocandins class of drugs was found active against most of the isolates of *C. auris*; however, echinocandin resistance in patients have been reported recently [8]. Researchers have reported multidrug resistance among *C. auris* isolates as a common phenomenon, severely restraining its treatment possibilities [6]. In South Africa, the first instance of infection caused by *C. auris* was reported in the 2014 and around 1,700 cases were detected between 2012 and 2016. Currently, *C. auris* is a widespread problem as it is found in almost 100 hospitals across South Africa with a vast majority of cases been reported from private hospitals in Gauteng Province. The underlying reason for rapid spread of *C. auris* infection is its ability to adhere to polymeric surfaces and biofilm formation [9].

Increasing prevalence of *C. auris* infection worldwide especially in South Africa motivated us to study pathogenic traits of this species. Farnesol, a first quorum sensing (QS) molecule identified in eukaryotic microorganisms [10], play an important role in an array of biological functions such as virulence, biofilm formation, and competence [11]. Numerous studies have reported the effect of farnesol on *C. albicans* growth and pathogenesis [12–14]. In *C. albicans*, farnesol inhibits the dimorphism [10] that prevent its establishment in different environmental conditions [15], it has antioxidant effects [16] and also inhibits drug transporters [17]. Additionally, farnesol showed low cytotoxicity without genotoxic effects [18]. Researchers have also suggested that farnesol competitively inhibits the efflux of antifungal compounds mediated by ABC drug transporters, possesses *in vitro* synergistic effect with various antifungals against single or mixed-species biofilm models and suppresses the resistance of *C. albicans* biofilms by regulating the expression of the gene *CYR1* and *PDE2* [19, 20]. With this background, we emphasized to study the effect of farnesol on growth, biofilms and reversal of drug resistance in different *C. auris* isolates.

## Methods

### Ethics statement

All the *C. auris* isolates (n = 25) were obtained from the Division of Mycology, National Institute of Communicable Diseases (NICD), Johannesburg, South Africa. To use these isolates in this study, an ethics waiver was obtained from the Human Research Ethics Committee of University of the Witwatersrand (M140159) and performed according to guidelines outlined in the Helsinki Declaration.

### *Candida* isolates

In this study, 25 *C. auris* (*C. auris* MRL 2397, *C. auris* MRL 2921, *C. auris* MRL 3499, *C. auris* MRL 3785, *C. auris* MRL 4000, *C. auris* MRL 4587, *C. auris* MRL 4888, *C. auris* MRL 5418, *C.*

*auris* MRL 5762, *C. auris* MRL 5765, *C. auris* MRL 6005, *C. auris* MRL 6015, *C. auris* MRL 6057, *C. auris* MRL 6059, *C. auris* MRL 6065, *C. auris* MRL 6125, *C. auris* MRL 6173, *C. auris* MRL 6183, *C. auris* MRL 6194, *C. auris* MRL 6277, *C. auris* MRL 6326, *C. auris* MRL 6333, *C. auris* MRL 6334, *C. auris* MRL 6338 and *C. auris* MRL 6339) and 1 *C. albicans* (SC5314) were used. All 25 clinical *C. auris* strains were from NICD's national active surveillance system for candidaemia (GERMS-SA) and the details about collection, identification and drug susceptibility can be obtained from van Schalkwyk and co-workers (2019) [21]. The isolates were stored in glycerol stock at -80˚C until required.

## Antifungal susceptibility profiles

The antifungal susceptibility profile of *C. auris* isolates were established by broth microdilution assay as per the recommended guidelines of Clinical and Laboratory Standards Institute (CLSI) reference document M27-A3 [22]. Briefly, inoculum was prepared by growing *C. auris* in Sabouraud Dextrose Broth (SDB) at 37˚C for 24h, centrifuged and resuspended yeast cells in fresh SDB and the turbidity of the suspension was adjusted to 0.5 McFarland Standard (equivalent to $5.0 \times 10^6$ CFU/ml) using MicroScan Turbidity meter (Beckman Coulter, CA, USA). Stock solutions of AmB and FLZ were prepared by using dimethyl sulfoxide (DMSO) and the range of concentrations tested were 16–0.004 µg/ml and 1000–0.25 µg/ml, respectively. Farnesol (Sigma Aldrich Co., USA) was obtained as a 3.7 M stock solution and was further diluted with 1% DMSO to achieve a concentration of 2000 mM. Two-fold dilutions of farnesol (100 µL) were prepared in 96-well flat-bottom microtiter plates, to obtain a concentration range of 500 to 0.24 mM after the inoculation of 100 µl of stock culture (0.5 McFarland) to each well. All the plates were incubated at 37˚C for 48h without shaking. In every set of experiment, cell free (sterility) and drug free (growth) controls were included for each *C. auris* isolates and all the isolates were tested in triplicate. *C. albicans* SC5314 was kept as a standard laboratory control in each test performed. Observation was made visually as well as by employing 3-(4,5-dimethyl-2-thiazolyl)-2,5-diphenyl-2H-tetrazolium bromide (MTT) reduction assay [23]. Briefly, a stock solution of MTT was prepared in Phosphate Buffer Saline (PBS) (5 mg/ml, filter sterilized and diluted 1:5 with pre-warmed sterile PBS). After 48h incubation, 50 µl of MTT solution was added to each well of the microtiter plate and was incubated for 5h at 37˚C. Subsequently, 100 µl of DMSO was added to solubilize the MTT-formazan product, which was measured at 490 nm by using a microplate reader (iMark, BioRad). MIC's were determined as the lowest concentration of the drugs (AmB and FLZ) and farnesol that inhibited growth of test organism.

## *Candida auris* biofilm formation

Biofilm formation by *Candida* spp. on medical devices is very common problem and life threatening for patients. Different clades of *C. auris* have also been reported to produce biofilm and therefore we studied the biofilm forming capability of *C. auris* strains isolated form different hospitals in South Africa. Biofilm formation by *C. auris* isolates (n = 25) was evaluated as described previously with slight modifications [24]. Briefly, 200 µl standardized cell suspension (0.5 McFarland) was inoculated in 96-well flat bottom microtiter plates at 37˚C for 2h. *C. albicans* SC5314 was used as a standard for biofilm formation. After incubation medium was aspirated and the wells were washed twice with sterile PBS, resulting in removal of planktonic cells. The wells of microtiter plates were reloaded with 200 µl of fresh medium and again incubated at 37˚C for 48h. Metabolic activities of biofilms formed by both the *Candida* species were measured using MTT reduction assay. Briefly, after biofilm formation 50 µl of 5mg/ml MTT solution was added to each well of the 96-well flat bottom microtiter plate and was

incubated at 37˚C for 5h. Subsequently, MTT was removed and 100 μl of DMSO was added to solubilize the MTT-formazan product. The resulting colored solution was quantified by measuring absorbance at 490 nm using a multi-well microplate reader (iMark, BioRad). The metabolic activities of biofilms formed by *C. auris* and *C. albicans* SC5314 was compared. The metabolic activity was also compared among *C. auris* isolates and those with higher readings were selected for further investigation (drug efflux and accumulation studies as well as for molecular analysis).

### Effect of farnesol on development of *C. auris* biofilms

To evaluate the activity of farnesol on sessile cells of *C. auris* and later development of biofilms, a method described previously was followed [23]. Briefly, to see the effect of farnesol on adherence and development of biofilms, freshly prepared *C. auris* cells (100 μl, 0.5 McFarland) were incubated with 100 μl farnesol (500–0.48 mM) (referred as zero time) in predetermined wells of 96-well flat bottom microtiter plate for initial 2h under biofilm forming conditions. Later on, planktonic cells were aspirated and wells were washed with sterile PBS followed by addition of fresh medium and incubation at 37˚C for 48h. Whereas, effect of farnesol on 4h premature *C. auris* biofilms was evaluated by inoculating 100 μl cell suspension (0.5 McFarland) into predetermined wells of 96-well flat bottom microtiter plates followed by incubation at 37˚C for 4h. After incubation the growth medium was removed followed by thorough washing with sterile PBS. After removal of non-adherent cells, different concentrations (500–0.48 mM) of farnesol were added to the wells of microtiter plate and incubated at 37˚C for 48h. Metabolic activities of the biofilms were measured using MTT reduction assay as described above.

### Effect of farnesol on mature biofilms

*C. auris* biofilms were allowed to grow at 37˚C for 12h and 24h under favorable biofilm forming conditions. The growth medium was removed, and biofilm was washed gently with sterile PBS. Farnesol (500–0.48 mM) was added to the predefined wells of microtiter plates and further incubated at 37˚C for 24h. The metabolic activity of treated and untreated biofilms was assessed by MTT reduction assay as described above.

The lowest concentration of farnesol where we reported ≥90% destruction in mature biofilm was recorded. Furthermore, biofilm inhibitory concentrations (BIC) were defined as the lowest concentration of farnesol where we report inhibition (≥90%) compared to the growth control.

### Confocal laser scanning microscopy (CLSM)

To further confirm the effect of farnesol on *C. auris* biofilm, CLSM was done. *C. auris* strain MRL5765 was allowed to grow on glass coverslips in 6-well microtiter plates under biofilm forming conditions. Farnesol (BIC) was administered in designated wells at different time points (4h, 12h and 24h) except the growth control wells (untreated cells). The plates were further incubated at 37˚C for 24h. Following incubation, the planktonic cells were aspirated and biofilms were gently washed twice with PBS and stained with fluorescent dye FUN-1 (Invitrogen, Thermo Fisher Scientific, South Africa) and concanavalin A (ConA)-Alexa Fluor 488 conjugate (Invitrogen, Thermo Fisher Scientific, South Africa). For staining, the coverslips were transferred to a new 6-well microtiter plate and incubated with 2 ml PBS containing FUN-1 (10 μM) and ConA-Alexa Fluor 488 conjugate (25 μg/ml) for 45 min at 37˚C in dark. FUN-1 (excitation = 543 nm and emission = 560 nm) is a vital dye and only live cells are capable of transporting it to the vacuole and result into orange-red cylindrical intra-vacuolar structures (CIVS) whereas in dead cells FUN-1 remain in the cytosol and fluoresces yellow-green [25].

ConA (excitation = 488 nm and emission = 505 nm) on the other hand fluoresces bright green when binds to α-mannopyranosyl and α-glucopyranosyl residues present in cell wall and biofilm matrix. After incubation with fluorescent dyes the glass coverslips were flipped on glass plates and stained biofilms were observed using a Zeiss Laser Scanning Confocal Microscope (LSM) 780 and Airyscan (Carl Zeiss). Multitrack mode was used to collect the images of green (ConA) and red (FUN-1) fluorescence simultaneously. The thickness or volume of whole biofilm was determined by collecting Z-stack picture and the distances between first and last fluorescent confocal plane was defined as biofilm thickness [26].

## Extracellular Rhodamine 6G efflux assay

Extracellular efflux of Rhodamine 6G (R6G) from *C. auris* cells were evaluated as described previously [27], with some minor adjustments. For this study four *C. auris* isolates (MRL 4000, MRL 5762, MRL 5765, and MRL 6057) were selected and *C. albicans* SC5314 was used as standard for efflux activity. These *C. auris* isolates were preferred over others because of higher biofilm forming capabilities as well as these isolates were resistant to FLZ and AmB. As these isolates were displaying comparatively higher virulence, they seem to be promising candidates for analyzing drug efflux pumps and to check effect of farnesol over these pumps. Briefly, *Candida* cells were grown on Sabouraud Dextrose Agar (SDA) plates at 37˚C for 24h. The standardized cell suspensions (0.5 McFarland) were inoculated in 50 ml growth media (SDB) at 37˚C for 8h. Post-incubation media was centrifuged (3,000 rpm for 5 min), washed twice with 25 ml PBS (without glucose) and was resuspended in the same buffer (2% cell suspension). The cells were then further incubated in 50 ml sterile PBS containing 2-deoxy-$_D$-glucose (5.0 mM) and 2,4 dinitrophenol (5.0 mM) for 45 min, resulting in de-energizing of cells. Followed by de-energization, the cells were washed and again resuspended in glucose-free PBS (0.5 McFarland). R6G (final concentration of 10 μM) was added to this resuspension and incubated for 40 min at 37˚C. After incubation cells were again washed and resuspended in glucose-free PBS and samples (2 ml) were withdrawn at definite intervals (0, 5, 10, 15, 20 min). After harvesting samples were pelleted at 3,000 rpm for 5 min and optical density of supernatant was recorded at 527 nm. To study the energy dependent R6G efflux, 0.1 M glucose was added after 20 min incubation to the cells resuspended in glucose-free PBS. The absorbance was recorded till 60 min of incubation with glucose and the last reading was recorded after overnight (20h) incubation. Positive as well as negative controls were included in all the experiments. The standard concentration curve of R6G was prepared for determining the actual concentration of R6G effluxed.

For competition assays, yeast cells were exposed for 2h to different concentration of farnesol (0.5 × MIC and MIC). Post exposure the cells were pelleted (3,000 rpm for 5 min) and washed twice with sterile PBS (without glucose). Thereafter, treated cells were de-energized and then equilibrated in R6G as stated above. Samples (2 ml) were withdrawn at predetermined time points (0, 5, 10, 15, 20 min), centrifuged (3,000 rpm for 5 min) and absorbance of supernatant was recorded at 527 nm. The estimation of energy dependent R6G efflux was done by adding 0.1 M glucose after 20 min incubation to the resuspended cells and reading were recorded till 60 min and last reading was recorded after 20h of incubation. Positive as well as negative controls were included in all the experiments. The standard concentration curve of R6G was prepared for determining the actual concentration of R6G effluxed.

## Intracellular Rhodamine 6G accumulation assay

Intracellular accumulation assay was executed as discussed earlier [27], with minor modification. Briefly, *C. auris* isolates (MRL 4000, MRL 5762, MRL 5765 and MRL 6057) cells were

grown overnight in SDB medium at 37°C. After incubation cells were centrifuged (3,000 rpm for 5 min) and washed twice in sterile PBS and re-inoculated in sterile SDB broth supplemented with farnesol (at $0.5 \times$ MIC and MIC) at 37°C for 2h. Post incubation, cells were pelleted (3,000 rpm for 5 min) and given sterile PBS wash. The washed cells were resuspended in sterile PBS (1.0 ml) supplemented with 2% glucose and 4 µM R6G and then incubated at 37°C for 30 min. Post incubation cells were washed twice with cold sterile PBS and the pellet was used for fluorescence microscopy.

## Real time PCR

Effect of farnesol on the expression of genes involved in biofilm formation and drug efflux pumps (Table 1) was evaluated by RT-qPCR. Briefly, *C. auris* (MRL 4000, MRL 5762, MRL 5765 and MRL 6057) cells at a concentration of $5 \times 10^6$ cells/ml were exposed to farnesol (MIC) followed by incubation at 37°C for 2h. Untreated cells were used as negative control. Total RNA was extracted using RNA MiniPrep kit (Inqaba Biotechnical Industries Ltd) following the manufacturer's instructions. Concentration of total RNA was measured using a Nanodrop 2000 spectrophotometer (Thermo Scientific, MA, USA). Purity of RNAs was assessed by determining $A_{260}/A_{280}$ ratio and a ratio above 2 was used for further experiment RT-qPCR. Thereafter, cDNA was synthesised using cDNA synthesis kit (Lasec South Africa Ltd) following manufacturer's instructions. Primers for the target and housekeeping genes (*ACT1*) were designed using online Primer3web version 4.1.0. (http://primer3.ut.ee/) (Table 1) and were synthesized from Metabion International AG, Germany (http://www.metabion.com/). RT-qPCR was performed using PowerUp™ SYBR™ Green Master Mix (Applied Biosystems) in a RocheLight® Cycler Nano instrument Real-time PCR system (Roche, Basel, Switzerland). The following thermal cycling conditions for all RT-qPCR reactions were used; UDG activation at 50°C for 2 min (Hold), Dual-lock DNA polymerase at 95°C for 2 min (Hold), 40 cycles of denaturation at 95°C for 15 sec, annealing at 53°C for 15 sec, and extension 72°C for 1 min. Dissociation curve conditions (melt curve stage) were as follows: Pre-melting at ramp rate of 1.6°C/sec, 95°C and 15 sec; Melting at ramp rate of 1.6°C/sec, 60°C and 1 min; Melting at ramp rate of 0.15°C/sec, 95°C and 15 sec. The dissociation curve and CT values were determined using the LightCycler Nano system. The gene expression was quantified and analyzed with respect to the housekeeping gene *ACT1* using formula $2^{-\Delta\Delta CT}$. The relative change in expression was estimated by normalizing to housekeeping gene (*ACT1*).

**Table 1. Nucleotide sequences for primers (5′—3′).**

| Gene | Forward primer | Reverse primer | Source |
|---|---|---|---|
| *CDR1* | GAAATCTTGCACTTCCAGCCC | CATCAAGCAAGTAGCCACCG | [28] |
| *CDR2* | GTCAACGGTAGCTGTGTG | GTCCCTCCACCGAGTATGG | [28] |
| *MDR1* | GAAGTATGATGGCGGGTG | CCCAAGAGAGACGAGCCC | [28] |
| *MDR2* | GGCGAGCTGTTGAGAATGTG | CTTCATGGCTTGCAACCTTC | [28] |
| *SNQ2* | ATCACCGAGGAATTGAGCAC | TCAACCTGTGAGCTTGATGC | This study |
| *HYR3* | CTGGTTTGACCTTCGTGGAT | GGCAGAGGTGACGTAGAAGC | This study |
| *IFF4* | AATGGTGCTGGTTGTGTGAA | AGTGAACCCAAGGTTGATGC | This study |
| *PGA26* | CCACGAACCTCCAAACAAGT | TGGTCACTGTGAGGGTGGTA | This study |
| *PGA52* | ACGAACACACCGTTGAATGA | AGTGCCATCTTGAGCGCTAT | This study |
| *PGA7* | GGCAGACTTTTCAGCTTTGG | AATCAATTTCCCGTTTGCAG | This study |
| *ALS5* | GCGATCCAATTTTGGAAGAA | GGTGCATCCCTATCTGAGGA | This study |
| *ACT1* | GAAGGAGATCACTGCTTTAGCC | GAGCCACCAATCCACACAG | [28] |

## Statistics

All the data and graphs were made and statistically analyzed using GraphPad prism version 5.01. All the experiments were carried out in triplicates, and the data obtained were presented as means ± standard error of the mean. Two-way ANOVA was used to compare untreated control with treated groups and P value less than 0.05 was considered significant.

## Results and discussion

### Antifungal susceptibility testing

All the clinical isolates of *C. auris* used in the present study were found sensitive to the farnesol within the MIC range of 62.5–125 mM. MIC values for AmB ranged from 0.125–4.0 μg/ml whereas for FLZ the MIC values ranged from 16–500 μg/ml (**Table 2**). CDC has established arbitrary breakpoints for *C. auris*, which were set at ≥32 μg/ml and ≥2 μg/ml for FLZ and AmB, respectively [9,29]. Based on these cutoff values, all the tested *C. auris* isolates except three (MRL 2397, MRL 3499, MRL 3785) were FLZ resistant whereas five *C. auris* isolates (MRL 2921, MRL 4000, MRL 5762, MRL 5765 and MRL 6057) were found resistant to AmB. Recent studies have also confirmed that *C. auris* isolates are usually resistant or less susceptible to azoles [30–32]. Furthermore, lower susceptibility of *C. auris* isolates against AmB is also in agreement with previous studies, where high AmB MICs for *C. auris* isolates were reported [33–35]. Inhibitory and modulatory effects of farnesol in *C. albicans* and other non-albicans *Candida* species has already being studied and its impact on biofilm formation, efflux pumps, and other virulence attributes is well established [12, 36–38]. However, this study for the first time reported inhibitory effect of farnesol on *C. auris* isolates. In comparison to other non-auris *Candida* species where low MICs for farnesol were reported, high concentrations are required to inhibit growth of *C. auris*. Farnesol at low concentration of 0.3 mM inhibits growth of *C. albicans* planktonic cells whereas, *C. dubliniensis* growth is inhibited at further lower concentration of 0.2 mM [39, 40]. As *C. auris* is MDR species, it was expected to have higher MIC values as compared to other *Candida* species.

### *C. auris* biofilm formation

Metabolic activity of untreated healthy biofilms was quantified by MTT and the results revealed that all the 25 isolates of *C. auris* were able to form biofilm. Whereas, the extent of biofilm formed varied among the isolates, only 16 out of 25 *C. auris* isolates were good biofilm

**Table 2. MIC values for AmB, FLZ and farnesol against isolates of *C. auris*.**

| *C. auris* isolates | AmB (μg/ml) | FLZ (μg/ml) | Farnesol (mM) | *C. auris* isolates | AmB (μg/ml) | FLZ (μg/ml) | Farnesol (mM) |
|---|---|---|---|---|---|---|---|
| MRL 2397 | 1.0 | 16.0 | 125.0 | MRL 6059 | 0.5 | 125.0 | 125.0 |
| MRL 2921 | 2.0 | 250.0 | 125.0 | MRL 6065 | 1.0 | 125.0 | 125.0 |
| MRL 3499 | 0.5 | 16.0 | 62.5 | MRL 6125 | 0.25 | 62.0 | 125.0 |
| MRL 3785 | 0.125 | 16.0 | 62.5 | MRL 6173 | 0.25 | 32.0 | 125.0 |
| MRL 4000 | 2.0 | 250.0 | 125.0 | MRL 6183 | 0.25 | 250.0 | 125.0 |
| MRL 4587 | 0.5 | 32.0 | 125.0 | MRL 6194 | 0.25 | 125.0 | 125.0 |
| MRL 4888 | 1.0 | 500.0 | 125.0 | MRL 6277 | 0.5 | 125.0 | 125.0 |
| MRL 5418 | 0.5 | 500.0 | 125.0 | MRL 6326 | 0.25 | 125.0 | 125.0 |
| MRL 5762 | 2.0 | 500.0 | 125.0 | MRL 6333 | 0.5 | 125.0 | 125.0 |
| MRL 5765 | 2.0 | 500.0 | 125.0 | MRL 6334 | 0.5 | 250.0 | 125.0 |
| MRL 6005 | 1.0 | 500.0 | 125.0 | MRL 6338 | 0.25 | 125.0 | 125.0 |
| MRL 6015 | 0.25 | 62.0 | 125.0 | MRL 6339 | 0.5 | 250.0 | 125.0 |
| MRL 6057 | 4.0 | 125.0 | 125.0 | | | | |

formers which was evident from higher MTT readings. Furthermore, biofilm formed by these 16 isolates were compared with biofilm formed by *C. albicans* SC5314. The metabolic activity recorded for *C. albicans* biofilm was 1.5 to 2.0 times higher than metabolic activity recorded for *C. auris* biofilm. Therefore, we concluded that *C. auris* isolates were having biofilm forming capability but less than *C. albicans* SC5314. When all 16 *C. auris* isolates were compared for their biofilm forming capability higher metabolic activity was recorded in MRL 4000, MRL 5762, MRL 5765 and MRL 6057; furthermore, these isolates were also found resistant to FLZ and AmB. Our results are in agreement with previous findings, which characterized and compared the biofilm formation of 16 different *C. auris* isolates with biofilms formed by *C. albicans* [32, 41]. In these studies, they reported *C. auris* biofilms are mainly composed of yeast cells whereas, extremely heterogeneous architecture was found in *C. albicans* biofilms. They also investigated and compared the virulence factors of *C. auris* isolates with *C. albicans*. Researchers also described and differentiated the biofilm forming ability of non-aggregative and aggregative strains of *C. auris* [41].

## Farnesol inhibits *C. auris* biofilm development and formation

Sixteen *C. auris* isolates showing good biofilm forming capability were selected for this study. Non adherent (pre-incubation/zero time) and adherent cell population (4h, 12h and 24h) of *C. auris* was treated with different concentrations of farnesol (500–0.48 mM) to determine its inhibitory effect over *C. auris* biofilm formation. The results demonstrated that incubation of isolates with the farnesol (125.0 mM) during initial phase of biofilm development stopped the cells from adhering and growing further, resulting in scarce or missing biofilms in the microtiter plate. Additionally, at concentration of 7.81 mM of farnesol we observed more than 50% inhibition in biofilm formation. When the cells were allowed to grow for 4h under biofilm forming conditions the biofilm inhibitory concentration (BIC) was found to be 125 mM. Furthermore, 12h and 24h mature biofilm became highly resistant to low concentration of farnesol and were inhibited at a high concentration of 500 mM. This was reflected clearly in the readings recorded at 490 nm, with lowest MTT readings at the highest concentration of farnesol. Percent inhibition of *C. auris* biofilms at different concentration is shown in **Fig 1**. The effect of farnesol on subsequent biofilm development decreased, as we allowed cells to adhere by increasing the incubation time. In comparison, good biofilm architecture was recorded for untreated biofilms (negative control), which was marked by readings obtained at 490 nm. The results further suggested that mature biofilms (12h and 24h) are more resistant. Although *C. auris* forms significantly reduced biofilms as compared to *C. albicans*, nevertheless, it has the ability to form biofilms on various medical devices [3]. In comparison to other non-auris *Candida* species where lower concentrations of farnesol were reported to have anti-biofilm effects, our results showed higher concentrations are required to inhibit biofilm formation and development. Sebaa and co-workers (2019) reported anti-biofilm activity of farnesol against *C. albicans* ATCC 10231 and 6 other strains isolated from dentures [42]. In the same study, it was reported that 3 mM farnesol exerted stronger action when added at the beginning of biofilm formation (>50% inhibition) than when added to 24h mature biofilms (<10% inhibition), and therefore concluded farnesol had a greater effect during the initial phases of biofilm formation. In another study, it was reported that metabolomic changes associated with hyphal suppression by farnesol in *C. albicans* reaches to complete hyphae inhibition at 1 mM concentration of farnesol [43]. Antifungal resistance and lower susceptibility among *C. auris* biofilms against commonly used drugs in comparison to *C. albicans* and *C. glabrata* has already been reported by researchers [41,44]. Results in the present study established the fact that farnesol prevented adherence of *C. auris* cells, which is a crucial step in biofilm formation.

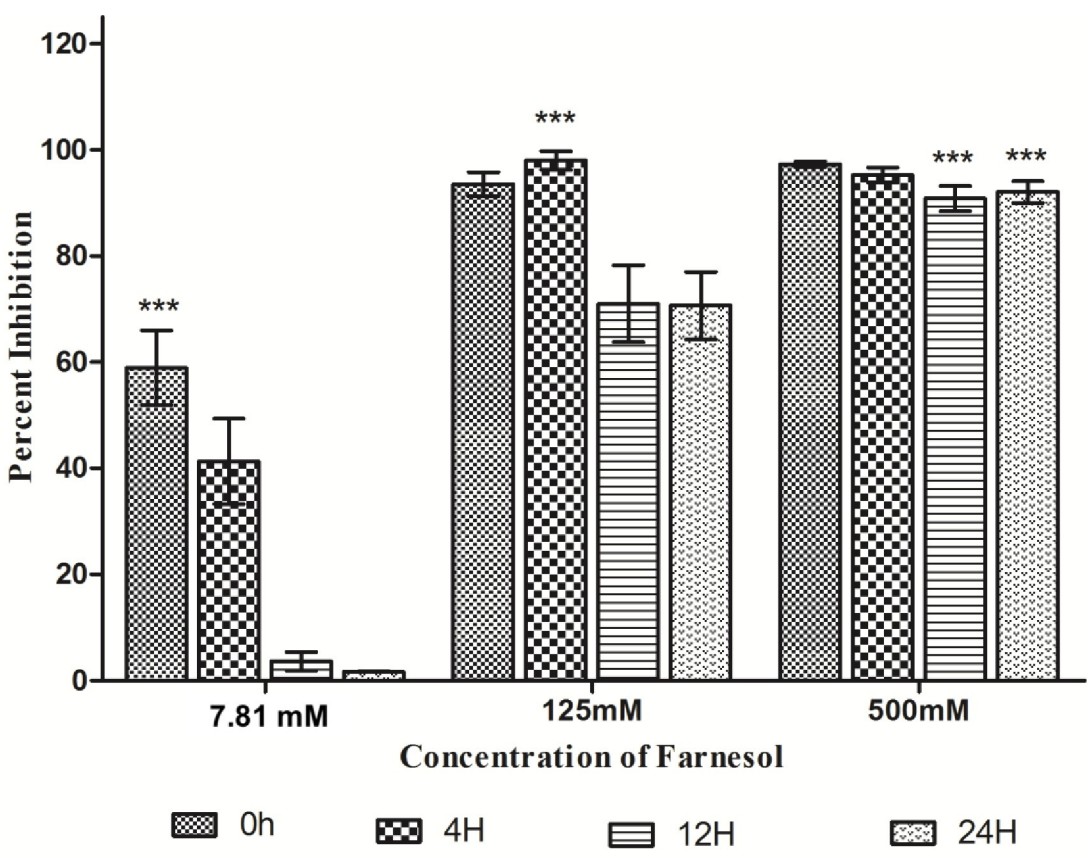

**Fig 1. Effect of farnesol on formation of *C. auris* biofilms.** *C. auris* cells were incubated under biofilm forming conditions with different concentrations of farnesol (500–0.48 mM) at time interval (pre-incubation, 4h, 12h, and 24h). Pre-incubation of *C. auris* cells with farnesol and treatment of *C. auris* biofilm formed after 4h, 12h and 24h of incubation with farnesol. MTT reduction assay was used for quantification. Two-way ANOVA was used to study statistical significance (***$P < 0.001$).

## CLSM analysis

*C. auris* biofilm architecture and antifungal activity of farnesol was examined by using CLSM technique. Biofilm cells at different growth phases were challenged with farnesol for 24h and stained with FUN-1 and ConA to evaluate the cell viability and thickness of biofilm extracellular matrix. The inhibitory effect of farnesol was assessed by comparing treated biofilms with untreated control under confocal microscope. CLSM analysis of untreated control revealed that as biofilm gets more mature their metabolic activity decreases, as seen by Fun-1 staining. Fun-1 permeabilizes into cells and only metabolically active cells convert Fun-1 to orange-red CIVS [25]. Highest metabolic activity was observed in 4h biofilms that was estimated by staining intensity of CIVS (orange-red colour) whereas as the staining decreased with increase in incubation time of biofilms (**Fig 2**). After 24 h of incubation under biofilm forming conditions, only a small subpopulation showed formation of CIVS, which was evident from orange-red fluorescence. These results are incongruent with the previous findings, where cells embedded in mature biofilm (48h) were reported to have reduced metabolic activity and limited growth as compared to cells with young biofilms (4h) [45]. Contrary to this with increasing age and maturity the thickness of extracellular matrix was found to increase which was evident by more green fluorescence during CLSM as a result of binding of ConA to glucose and mannose

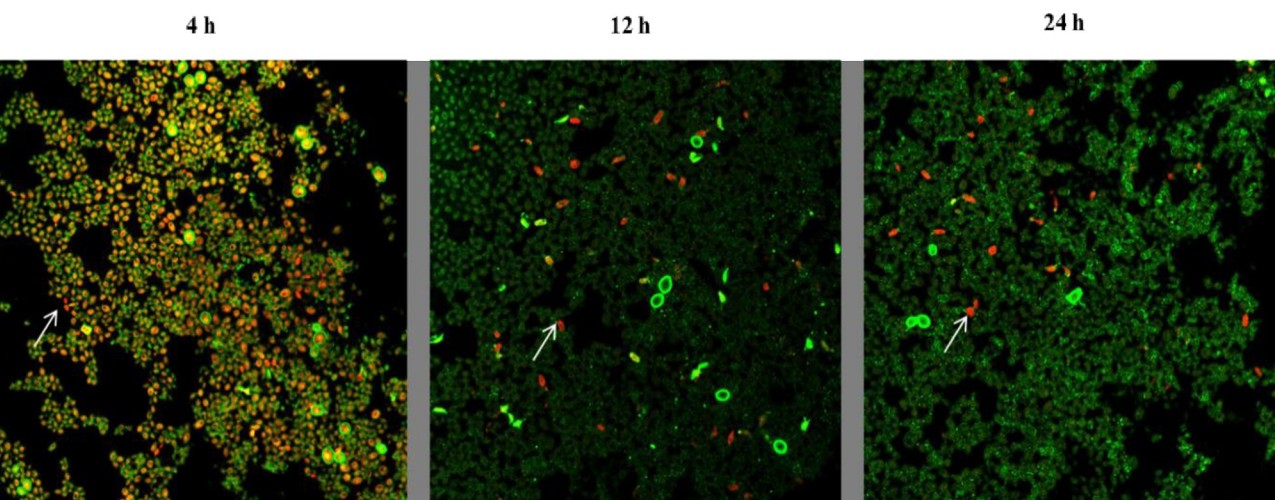

**Fig 2. Metabolic activity of *C. auris* cells in biofilm decreases with increasing maturity.** *C. auris* strain MRL 5765 cells were stained with FUN-1 and ConA- Alexa Fluor 488 conjugate stain after incubation of biofilm for 4h, 12h and 24h respectively. The image utilizes a 63X oil immersion objective with magnification, X1. The cells that were live and active produce orange-red cylindrical intra-vacuolar structures (CIVS). The increase in green fluorescence shows the increase in biofilm extracellular matrix with increasing incubation time.

residues present in cell wall and biofilm matrix. The thickness of 4h young, 12h and 24h mature biofilm was found to be around 9.08 μm, 12.11 μm and 13.93 μm respectively.

In our current study, CLSM revealed that after treatment with farnesol (BIC; 125 mM for 4h and 500 mM for 12h and 24h) the biofilms were abrogated and the viability of cells in the biofilm was altered, which was evident by aggregate of cells scattered on the coverslip and presence of yellow-green structures showing metabolically inactive cells (**Fig 3**). Furthermore, farnesol also decrease the whole volume of the biofilm, the thickness of treated 4h, 12h, and 24h biofilms were measured and found to be 8.48 μm in case of treated 4h, and 12h biofilms whereas the thickness of treated 24 h biofilm was recorded as 9.69 μm. It is clear that farnesol significantly affected the sustainability of yeast cells in biofilms pointing towards the ability of farnesol to penetrate cell membrane resulting in an effective anti-biofilm activity.

## Farnesol inhibits R6G efflux mediated transporters

MDR efflux pumps contributes to drug resistance and their functional activity *in vitro* can be studied by using R6G, which is a recognized efflux substrate [46]. In FLZ non-responsive *Candida* cells ATP-dependent efflux pumps (ABC superfamily) throws out R6G dye after entering cells passively [47]. In this study, we are measuring the inhibitory/modulatory outcome on MDR efflux transporters of *C. auris*. Here we investigated four isolates of *C. auris* (MRL 4000, MRL 5762, MRL 5765 and MRL 6057) for the passage of substrate, R6G (10 μM) after manifestation of farnesol (0.5 × MIC and MIC). When the cells were incubated with PBS (without glucose) we noted an instant uptake of R6G dye by the cells, which came to stability after 30 min. After addition of glucose, control *Candida* cells (untreated) showed energy and time dependent efflux of R6G dye from the cells as they are rich in membrane bound MDR transporter pumps. This was clearly evident from steady rise in the concentration of R6G dye extracellularly (**Fig 4**). Whereas, farnesol (0.5 × MIC) resulted in inhibition of energy dependent R6G efflux from treated cells confirming its role in modulating/inhibiting MDR efflux transporters. Higher concentration of farnesol (MIC) also had complete inhibitory effect on these energy dependent efflux pumps.

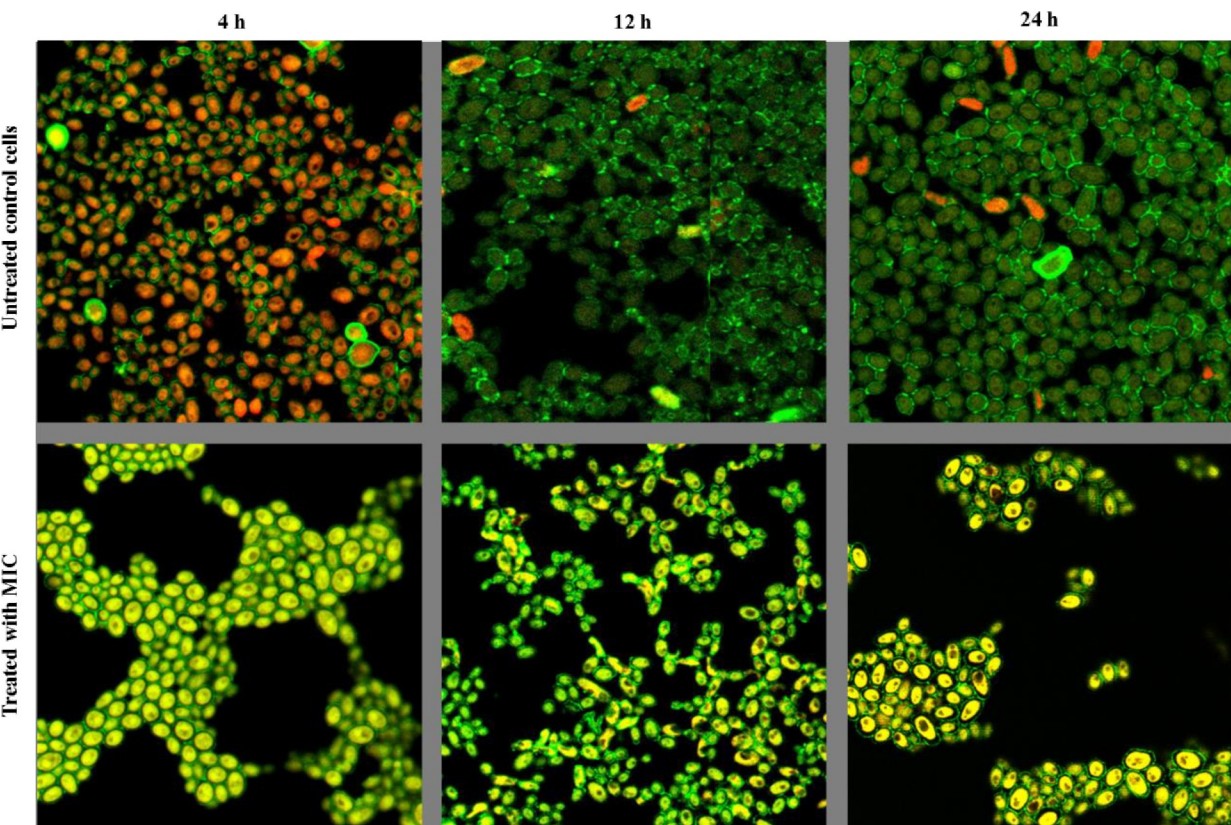

**Fig 3. Inhibitory effect of farnesol on *C. auris* biofilms.** CLSM analysis of *C. auris* strain MRL 5765 biofilms treated with farnesol at different stages of growth and development (4h, 12h and 24h). The image utilizes ConA- Alexa Fluor 488 conjugate and FUN-1 staining, a 63X oil immersion objective with magnification, X2.5. In above images structures shown in green (ConA) represent the fungal cell wall and biofilm matrix; structures in yellow-green (FUN-1) are metabolically inactive and dead cells; structures in orange-red (FUN-1) are the viable cells.

Inhibition of efflux pumps in *C. albicans* by phytochemicals has been well established by the researchers [27, 48]. Researchers have also explained the role of MDR efflux transporters (ABC and MFS transporter) in resistance and the genes encoding these pumps in *C. auris* [44]. Our results are in agreement with previous studies [49], who reported high activity of ABC-type efflux in *C. auris* than *C. glabrata*, isolates proposing *C. auris* with efflux-mediated intrinsic resistance to azoles. It has been reported by authors that in *C. albicans* farnesol particularly targets CaCdr1p and CaCdr2p (ABC transporters), resulting in changes in resistance to azoles in *C. albicans* isolates but in a concentration dependent manner [37]. Interestingly, kinetics data cleared that farnesol results in competitive inhibition with an unchanged $V_{max}$ and higher $K_m$ values in cells overexpressing CaCdr1p [17].

To the best of our knowledge we are exploring outcome of farnesol on MDR efflux transporters in *C. auris* isolates for the first time. Our results clearly state that farnesol is completely inhibiting ABC MDR transporters similar to that reported in *C. albicans*.

## Farnesol inhibits R6G extrusion

To further confirm the effect of farnesol on activity of *C. auris* efflux pumps, accumulation of R6G dye inside the cells was studied. Untreated *C. auris* isolates were used as a control for estimating actual accumulation of R6G in cells. The fluorescence was recorded high in case of farnesol (0.5 × MIC and MIC) treated *C. auris* cells, suggesting more R6G accumulation intracellularly in comparison with untreated control cells (**Fig 5**).

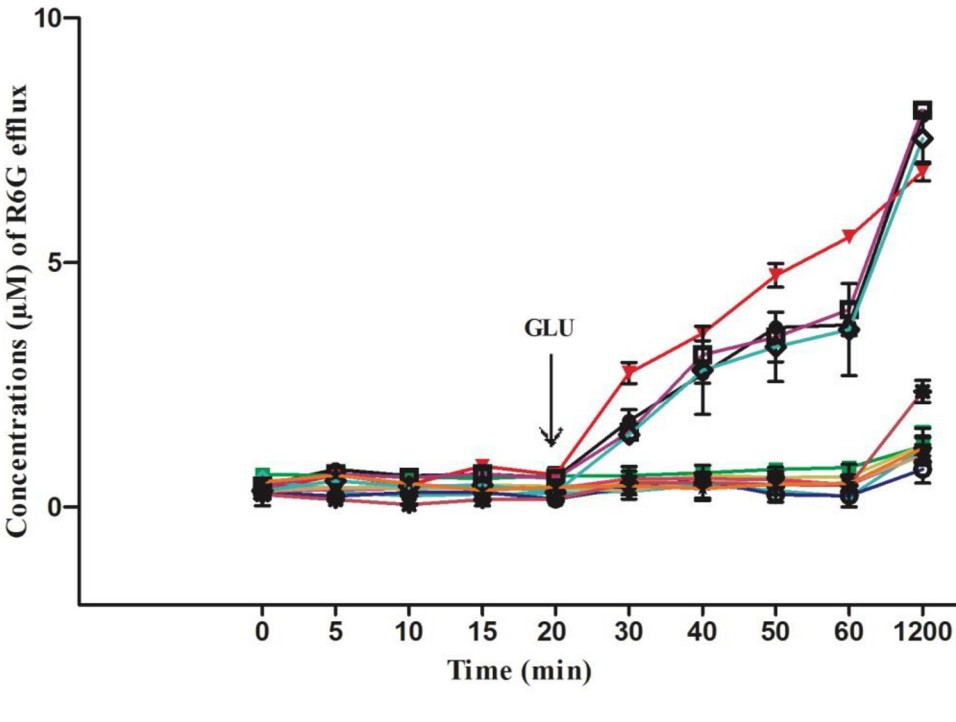

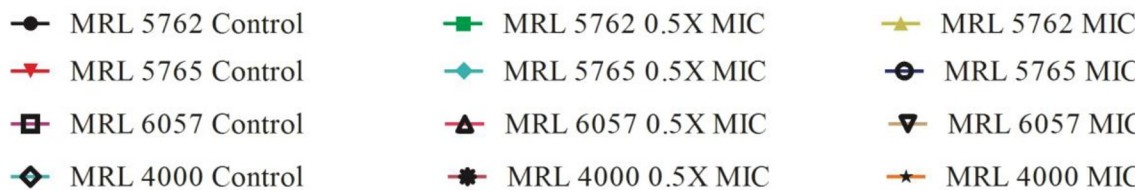

**Fig 4. Effect of farnesol on extracellular R6G efflux.** Figure showing concentration of extracellular R6G in untreated control and treated *C. auris* cells. Addition of glucose (0.1 M; indicated by an arrow) activated R6G transport from the cells and absorbance was recorded at 527 nm. Concentration of extracellular R6G was determined by standard curve. The values presented here is average of three independent experiments.

*Candida auris* is known to be highly multidrug resistant and among the various resistance mechanisms, the most dominant approach for throwing drug out of cell is the overexpression of MDR transporters and makes *Candida* unresponsive to antifungal therapies. The most common reason for MDR is efflux transporters from ABC superfamily, such as CaCdr1p and CaCdr2p [50, 51]. Despite several studies reporting the inhibition of efflux pumps in *C. albicans*, there are no reports on inhibition of efflux transporters in *C. auris* till date. Whereas, lot of work have been published on inhibition of this transporters in *C. albicans* and it has been published that farnesol effectively inhibits/modulates ABC MDR pumps (CaCdr1p and CaCdr2p) and did not affect much in case of MFS efflux pumps (CaMdr1p) [17, 37]. This study for the first-time reported efficacy of farnesol to block efflux pumps in *C. auris* isolates.

## Farnesol results in reduced expression of genes associated to resistance

*Candida auris* is an emerging MDR pathogen with higher MIC values against commonly used antifungal drugs when compared to other *Candida* species and the molecular basis of its MDR still remains unclear. Few studies have come up recently that has provided insight into

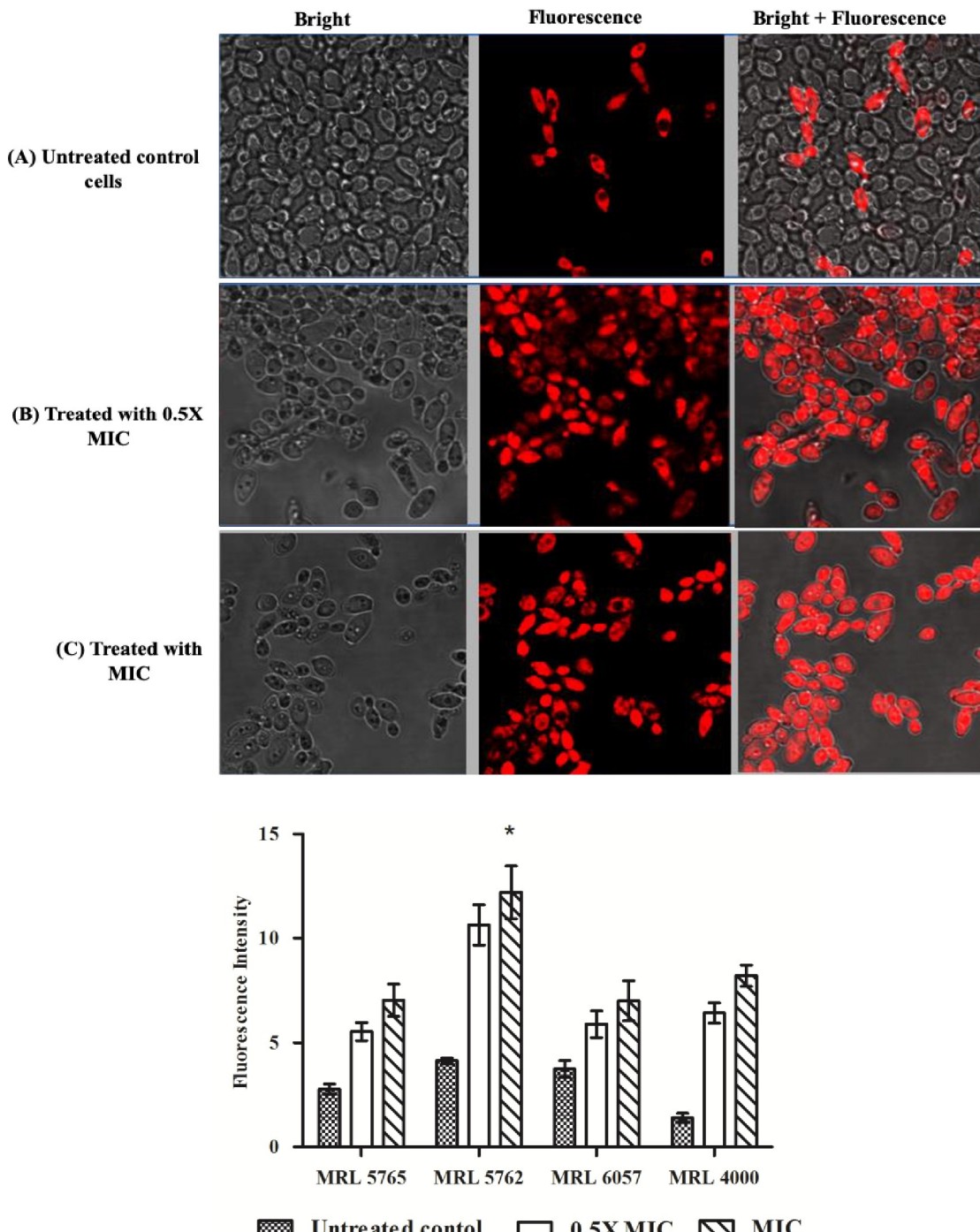

**Fig 5. R6G intracellular accumulation.** Fluorescence images of R6G stained *C. auris* MRL 5765 cells. The isolate was observed under bright field and fluorescence microscopy. (A) The figure shows untreated *C. auris* cells that have effluxes R6G dye during incubation with glucose and only few cells retain R6G fluorescence. Treatment with $0.5 \times$ MIC (B) and MIC (C) results in accumulation of R6G inside the cells after incubation with glucose which is clearly represented by high fluorescence inside the cells. (D) Fluorescence intensity is represented in the form of bar graph and was quantified with the help of Image J software.

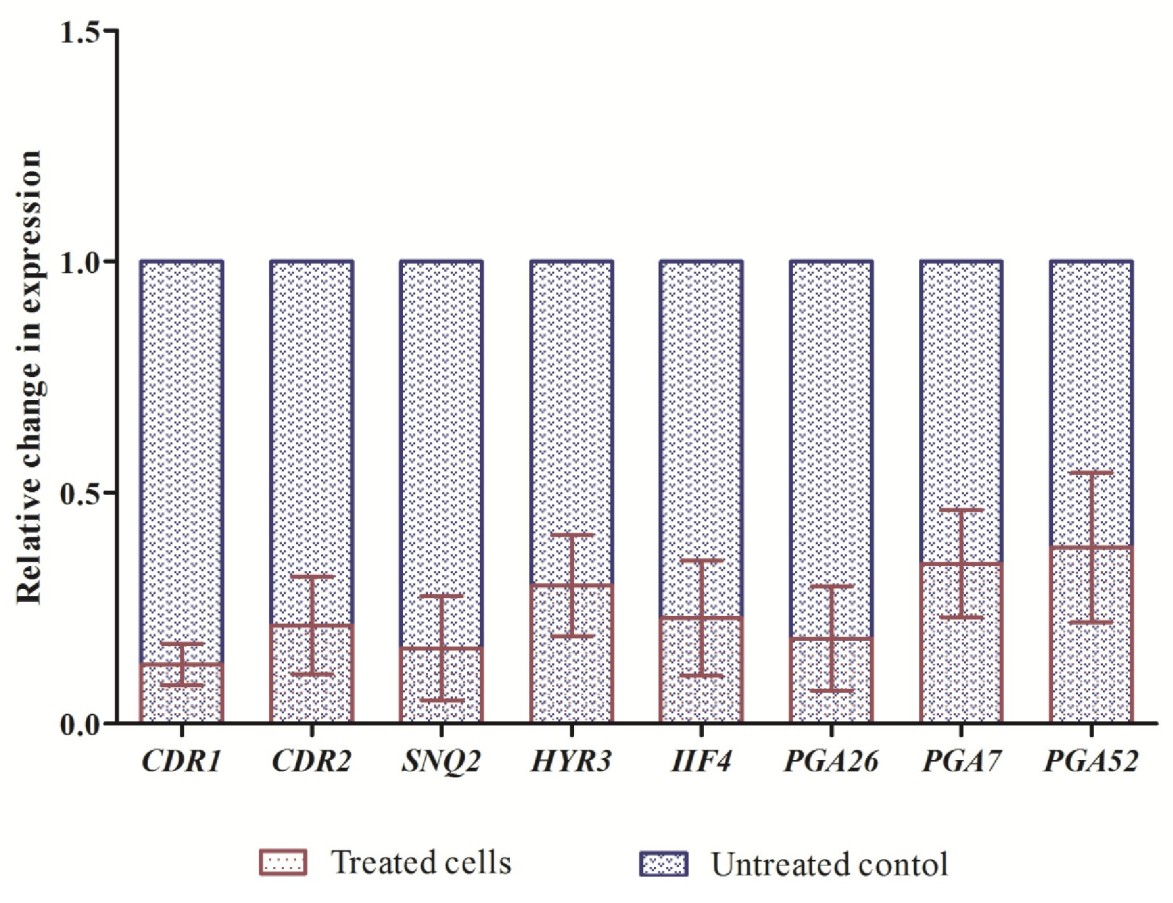

**Fig 6. Comparative variation in gene expression of *C. auris* isolates, *CDR1*, *CDR2*, *SNQ2*, *HYR3*, *IIF4*, *PGA26*, *PGA7* and *PGA52*
determined by RT-qPCR.** Data represents fold change in gene expression by adopting comparative Ct method. The gene expression between
control and treated were compared and was normalized to a value of 1 and *ACT*1 was used as housekeeping gene.

transcriptomic assembly of *C. auris* and revealed the possible mechanism of drug resistance
among these isolates [28, 44]. For understanding the molecular basis of farnesol on *C. auris*
drug efflux pumps and biofilm formation, we studied gene expression in *C. auris*. *IIF4*, *PGA26*,
*PGA7*, *PGA52* and *HYR3* genes played an active role in adhesion of biofilms and *SNQ2*, *CDR1*,
*CDR2*, *MDR1*, and *MDR2* genes were functionally related to efflux pumps in *C. auris* [44]. After
the active role of farnesol on the biofilm and efflux pumps, we studied the effect of farnesol on
the expression levels of these genes in different *C. auris* isolates. After treatment with 125 mM
farnesol, all the above-mentioned genes were significantly downregulated at varying levels (**Fig
6**). Based on these results, we therefore speculated that the inhibition of drug efflux pumps in all
the tested isolates by farnesol may be related to downregulation of genes related to efflux pump
in *C. auris* (*CDR1*, *CDR2* and *SNQ2*). The downregulation of *CDR1* and *CDR2* genes after treat-
ment with farnesol confirms its potential to inhibit ABC drug efflux transporters. *MDR1* and
*MDR2*, on the other hand, were not having consistent expression level in all the isolates of *C.
auris*. It has already been mentioned by researchers that *CDR1* is relatively more expressed as
compared to *MDR1* [28]. Additionally, both *CDR2* and *MDR2* expression were not raised in
any of the isolates. However, in our study both *CDR1* and *CDR2* are over expressed than *MDR1*

and *MDR2*. In addition, downregulation of genes (*IIF4*, *PGA26*, *PGA7* and *PGA52*) associated to adhesion also supports our *in vitro* biofilm inhibition results. Interestingly, *ALS5* did not expressed in any of the four isolates, which could be due to its late upregulation in the latter stages of mature biofilms or due to ALS-independent adherence mechanism in *C. auris* [44]. However, further in-depth studies are required to prove these claims.

Our results showed that farnesol, a quorum sensing molecule, has ability to inhibit growth of multidrug resistant *C. auris* isolates and also has capacity to suppress *C. auris* biofilm formation and efflux pumps. In addition, the effect of farnesol was related to the molecular mechanisms as all the biofilm and efflux pump associated genes were downregulated at varying levels. Furthermore, farnesol has a property to modulate the function of ABC efflux transporters of *C. auris* and thereby can be utilized in reversing azole drug resistance in these isolates. To the best of our knowledge, this is the first study exploring modulatory outcome of farnesol on development of biofilms and efflux pumps of *C. auris* globally and specifically to South African clade, it will be helpful in unrevealing possible mechanisms of action of farnesol against biofilm formation and efflux transporters of *C. auris*.

## Acknowledgments

Authors are grateful to Prof Nelesh Govender for providing *C. auris* strains used in this study. The CLSM facility provided by Life Sciences Imaging Facility (LSIF) at Faculty of Health Sciences, University of the Witwatersrand is duly acknowledged by authors.

## Author Contributions

**Conceptualization:** Vartika Srivastava, Aijaz Ahmad.

**Data curation:** Vartika Srivastava.

**Formal analysis:** Vartika Srivastava, Aijaz Ahmad.

**Funding acquisition:** Aijaz Ahmad.

**Methodology:** Vartika Srivastava, Aijaz Ahmad.

**Project administration:** Aijaz Ahmad.

**Supervision:** Aijaz Ahmad.

**Validation:** Aijaz Ahmad.

**Writing – original draft:** Vartika Srivastava.

**Writing – review & editing:** Aijaz Ahmad.

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
