## [Decision Letter · Decision Letter 0]

24 Feb 2020

PONE-D-19-35054

Abrogation of pathogenic attributes in drug resistant Candida auris strains by farnesol

PLOS ONE

Dear Dr. Ahmad

Thank you for submitting your manuscript to PLOS ONE. After careful consideration, we feel that it has merit but does not fully meet PLOS ONE’s publication criteria as it currently stands. Therefore, we invite you to submit a revised version of the manuscript that addresses the points raised during the review process.

Due to the importance of the multi-drug resistant strain Candida auris, this manuscript is of interest. However, there are some issues that need to be addressed. First the material and methods re strains and PCR need to be expanded upon. Since C. auris is reported to demonstrate more MDR than other strains, there should be a short discussion on the relative sensitivity between C. auris and other Candida strains to farnesol. Finally both reviewers have noted other minor points that should be addressed. Please address both reviewers.

We would appreciate receiving your revised manuscript by Apr 09 2020 11:59PM. To enhance the reproducibility of your results, we recommend that if applicable you deposit your laboratory protocols in protocols.io, where a protocol can be assigned its own identifier (DOI) such that it can be cited independently in the future. For instructions see: http://journals.plos.org/plosone/s/submission-guidelines#loc-laboratory-protocols

We look forward to receiving your revised manuscript.

Kind regards,

Joy Sturtevant

Academic Editor

PLOS ONE

Journal Requirements:

Please ensure that your manuscript meets PLOS ONE's style requirements, including those for file naming. The PLOS ONE style templates can be found at http://www.plosone.org/attachments/PLOSOne_formatting_sample_main_body.pdf and http://www.plosone.org/attachments/PLOSOne_formatting_sample_title_authors_affiliations.pdf

Reviewers' comments:

Reviewer's Responses to Questions

**Comments to the Author**

1. Is the manuscript technically sound, and do the data support the conclusions?

Reviewer #1: Partly

Reviewer #2: Yes

2. Has the statistical analysis been performed appropriately and rigorously? 

Reviewer #1: I Don't Know

Reviewer #2: Yes

3. Have the authors made all data underlying the findings in their manuscript fully available?

Reviewer #1: Yes

Reviewer #2: Yes

4. Is the manuscript presented in an intelligible fashion and written in standard English?

Reviewer #1: No

Reviewer #2: Yes

5. Review Comments to the Author

Reviewer #1: Manuscript describes the impact of farnesol on Candida auris strains, believed to be all belonging to the South African-clade as they are all from South Africa (but there is no proper ID provided, so some isolates can be from Asia or South America). A subset is chosen, but it is unclear on which grounds, to run the experiments on. The manuscript itself seems be hastily written and this needs more attention. Another flaw is that statements are made that (for example) a RT-qPCR method for the expression of a large number of genes is used as published before... but only part of the genes were used in the previously published study. On the next page authors provide primers for the missing genes, but this is not a proper Material & Methods description. For most of the experiments a cell suspension is used, it is unknown how the authors counted the cells and if this is based on a McFarland range or that a cell counter was used. See more comments in the attached file (up to page 14 as the reminder of the document does not contain that many comments; except that the references needs to be checked to be in the correct style).

Reviewer #2: This study investigates the effect of a quorum-sensing molecule (farnesol) on the emerging yeast Candida auris, which is particularly virulent and often resistant to the usual antifungal agents. For these reasons, the data reported in this study is worth considering for publication in PLOS ONE. The pre-print of this manuscript has been deposited on the BioRxiv server.

The submitted manuscript is well written. The authors have carefully chosen different techniques to analyze the effect of the quorum-sensing molecule (farnesol) on C. auris. The authors have combined the description and interpretation of the results for each of the techniques considered. This makes the manuscript clear and easy to read. However, a general discussion/conclusion would be essential to discuss the sensitivity of C. auris to farnesol observed in this study, with the data available in the literature for other Candida’s. Thus, the authors could emphasize that their strains of C. auris sometimes present a higher MIC for farnesol than that observed by other authors in other species of Candida.

Below you will find some remarks that should be clarified for a better reading/understanding.

1 °) Introduction and/or general conclusion. Authors should justify the interest of farnesol to inhibit the growth of C. auris compared to other quorum-sensing molecules (such as tyrosol) or compared to molecules belonging to other physiological mechanisms of defense.

2 °) Material and methods. C. albicans SC5314 is well known as a reference strain from a library. The authors should further specify the status of their C. auris strains: reference strains from a bacterial library or wild strains well characterized by a National Institute? If the authors consider the strains studied as isolates from their countries, is it worth mentioning the internal laboratory identification number? This could contribute to lightening the text.

3 °) Material and methods. Lines 101-103: The origin of farnesol should be mentioned as well as the concentration of the stock solution. Are farnesol concentrations (from 250 to 0.48 mM) indicative of reactant concentration or final concentration in the assay tube/well? Line 132: "500 to 0.488 mM “ rather than "500 mM to 0.488 mM”.

Minor remarks.

Line 55-56: Abbreviations of amphotericin (AmB) and fluconazole (FLZ) should be clearly indicated in parentheses after the first mention of each in the text. “…resistant to azoles as fluconazole (FLZ) and …unaffected to polyenes as amphotericin (AmB). …”

Line 114: Revise the sentence: “… the lowest concentration of (?) AmB farnesol that …

Line 192: Choose 20h or 1200 min: only one of both is enough.

Line 290: Clarify “pre-incubation (0h)” statement. Does it well mean the absence of pre-incubation?

6. PLOS authors have the option to publish the peer review history of their article (what does this mean?). If published, this will include your full peer review and any attached files.

Reviewer #1: No

Reviewer #2: No

---

## [Author Response · Author response to Decision Letter 0]

8 Apr 2020

Response to Reviewers

Editor comments: 

Comment 1: First the material and methods re strains and PCR need to be expanded upon

Response: As suggested material and methods section has been expanded with more information on strains used. All the strains are placed in numerical order in text. Methodology for antifungal susceptibility, biofilm assays, R6G efflux and Real Time PCR has been detailed in the revised manuscript. PCR methodology has also been extended with emphasis over the source from where gene sequences were obtained for the present study and the basis of gene selection.

Comment 2: Since C. auris is reported to demonstrate more MDR than other strains, there should be a short discussion on the relative sensitivity between C. auris and other Candida strains to farnesol.

Response: Page 14, Line 285-290: Further discussion comparing the effect of farnesol on MDR C. auris with other Candida species have been included in the revised results and discussion section. 

Reviewer #1: 

Comment 1: All strains belonging to the South African-clade as they are all from South Africa (but there is no proper ID provided, so some isolates can be from Asia or South America). 

Response: Page 5, Line 96-100: All the isolates used in this study were obtained from Division of Mycology, National Institute of Communicable Diseases (NICD), Johannesburg, South Africa. All these isolates were from NICD’s national active surveillance system for candidaemia (GERMS-SA) and their details are available online. All the isolates were collected from different hospitals in South Africa and have been classified into South African-clade C. auris. All the IDs used in the current study are based on the IDs provided by the Mycology reference laboratory of NICD. These details have now been provided in the revised manuscript. 

Comment 2: A subset is chosen, but it is unclear on which grounds, to run the experiments on.

Response: Page 9-10, Line 198-201: C. auris isolates, MRL 4000, MRL 5762, MRL 5765 and MRL 6057 were used as a subset to run the experiments because these isolates were best biofilm formers among all the 25 C. auris isolates and they were also found resistant to FLZ and AmB. Therefore, these isolates were selected for understanding activity of drug efflux pumps and expression of biofilm and resistance associated genes in C. auris. As well as effect of farnesol on virulence associated traits of C. auris.

Comment 3: The manuscript itself seems be hastily written and this needs more attention. Another flaw is that statements are made that (for example) a RT-qPCR method for the expression of a large number of genes is used as published before... but only part of the genes were used in the previously published study. On the next page authors provide primers for the missing genes, but this is not a proper Material & Methods description.

Response: The whole manuscript has been edited for the major and minor mistakes. The sections have been detailed where required. Materials and methods section have been revised in general and for Real Time PCR section in particular. Details related to use of primers has been cleared in the revised manuscript. 

Comment 4: For most of the experiments a cell suspension is used, it is unknown how the authors counted the cells and if this is based on a McFarland range or that a cell counter was used.

Response: Cell suspension was adjusted to 0.5 McFarland Standard (equivalent to 1.0 – 5.0 � 106 CFU/ml) using MicroScan Turbidity meter. This has now been mentioned in section Antifungal susceptibility profiles.

Comment 5: See more comments in the attached file (up to page 14 as the reminder of the document does not contain that many comments; except that the references needs to be checked to be in the correct style).

Response: Authors are thankful to the reviewer for all the mentioned corrections and suggestions, which has now been incorporated in the revised manuscript. 

Reviewer #2:

This study investigates the effect of a quorum-sensing molecule (farnesol) on the emerging yeast Candida auris, which is particularly virulent and often resistant to the usual antifungal agents. For these reasons, the data reported in this study is worth considering for publication in PLOS ONE. The pre-print of this manuscript has been deposited on the BioRxiv server.

The submitted manuscript is well written. The authors have carefully chosen different techniques to analyze the effect of the quorum-sensing molecule (farnesol) on C. auris. The authors have combined the description and interpretation of the results for each of the techniques considered. This makes the manuscript clear and easy to read.

Response: We are thankful to reviewer for his thorough review of the article and for the valuable comment.

Comment 1: However, a general discussion/conclusion would be essential to discuss the sensitivity of C. auris to farnesol observed in this study, with the data available in the literature for other Candida’s. Thus, the authors could emphasize that their strains of C. auris sometimes present a higher MIC for farnesol than that observed by other authors in other species of Candida.

Response: Page 14, Line 285-290: Further discussion comparing the effect of farnesol on C. auris with other Candida species have been included in the revised results and discussion section. 

Comment 2: Introduction and/or general conclusion. Authors should justify the interest of farnesol to inhibit the growth of C. auris compared to other quorum-sensing molecules (such as tyrosol) or compared to molecules belonging to other physiological mechanisms of defense.

Response: Page 4, Line 74-87: This study is focused on farnesol than any other quorum sensing molecule based on the previous literature reporting the effect of farnesol on growth, biofilms and reversal of drug resistance in different non-auris Candida isolates. The relevance of use of farnesol is detailed in the revised introduction. 

Comment 3: Material and methods. C. albicans SC5314 is well known as a reference strain from a library. The authors should further specify the status of their C. auris strains: reference strains from a bacterial library or wild strains well characterized by a National Institute? If the authors consider the strains studied as isolates from their countries, is it worth mentioning the internal laboratory identification number? This could contribute to lightening the text.

Response: All the isolates used in this study were obtained from Division of Mycology, National Institute of Communicable Diseases (NICD), Johannesburg, South Africa. All these isolates were from NICD’s national active surveillance system for candidaemia (GERMS-SA) and their details are available online. All the isolates were collected from different hospitals in South Africa and have been classified into South African-clade C. auris. All the IDs used in the current study are based on the IDs provided by the Mycology reference laboratory of NICD. These details have now been provided in the revised manuscript. 

Comment 4: Material and methods. Lines 101-103: The origin of farnesol should be mentioned as well as the concentration of the stock solution. Are farnesol concentrations (from 250 to 0.48 mM) indicative of reactant concentration or final concentration in the assay tube/well? Line 132: "500 to 0.488 mM “ rather than "500 mM to 0.488 mM”.

Response: Line no: 131-135 “ Farnesol was purchased from Sigma Aldrich Co., USA with a stock concentration of 3.7 M solution, which was further diluted with 1% DMSO to achieve a concentration of 2000 mM which was added in the first column of 96 well plate for serial dilution. The concentration of farnesol in wells of 96-well flat bottom microtiter plate ranged from 500 mM to 0.24 mM. All the details and corrections has been included in the revised methodology section.

"500mM to 0.488 mM has been corrected to "500 to 0.488 mM” in the revised manuscript.

Minor remarks.

Comment 1: Line 55-56: Abbreviations of amphotericin (AmB) and fluconazole (FLZ) should be clearly indicated in parentheses after the first mention of each in the text. “resistant to azoles as fluconazole (FLZ) and …unaffected to polyenes as amphotericin (AmB)”

Response: All the abbreviations have now been clearly indicated in parentheses after the first mention in the text.

Comment 2: Line 114: Revise the sentence: “… the lowest concentration of (?) AmB farnesol that

Response: Sentence has been clearly rewritten to avoid confusion. 

Comment 3: Line 192: Choose 20h or 1200 min: only one of both is enough.

Response: 20h in place of 1200 min has been used consistently throughout the revised manuscript.

Comment 4: Line 290: Clarify “pre-incubation (0h)” statement. Does it well mean the absence of pre-incubation?

Response: To avoid confusion, “(0h)” has been reworded to “zero time point” in revised manuscript. To evaluate the effect of farnesol on adherence and development of biofilms, C. auris cells were pre-incubated with different concentrations of farnesol (referred as zero time/0h) for initial 2h in predetermined wells of microtiter plate under biofilm forming conditions. Later on, planktonic cells were removed and wells were washed with sterile PBS followed by addition of fresh medium and incubation at 37°C for 48h. For treated group, cells were pre-incubated (2h) with farnesol under favourable biofilm forming conditions (to check effect on adherence and formation of biofilm) whereas untreated cells were pre-incubated without farnesol for 2h (for proper adherence to surface).

This has now been detailed in the methodology section in the revised manuscript.

---

## [Decision Letter · Decision Letter 1]

29 Apr 2020

Abrogation of pathogenic attributes in drug resistant Candida auris strains by farnesol

PONE-D-19-35054R1

Dear Dr. Ahmad,

We are pleased to inform you that your manuscript has been judged scientifically suitable for publication and will be formally accepted for publication once it complies with all outstanding technical requirements.

With kind regards,

Joy Sturtevant

Academic Editor

PLOS ONE

Additional Editor Comments (optional):

Reviewers' comments:

Reviewer's Responses to Questions

**Comments to the Author**

1. If the authors have adequately addressed your comments raised in a previous round of review and you feel that this manuscript is now acceptable for publication, you may indicate that here to bypass the “Comments to the Author” section, enter your conflict of interest statement in the “Confidential to Editor” section, and submit your "Accept" recommendation.

Reviewer #1: All comments have been addressed

Reviewer #2: All comments have been addressed

2. Is the manuscript technically sound, and do the data support the conclusions?

Reviewer #1: Yes

Reviewer #2: (No Response)

3. Has the statistical analysis been performed appropriately and rigorously? 

Reviewer #1: I Don't Know

Reviewer #2: (No Response)

4. Have the authors made all data underlying the findings in their manuscript fully available?

Reviewer #1: Yes

Reviewer #2: (No Response)

5. Is the manuscript presented in an intelligible fashion and written in standard English?

Reviewer #1: Yes

Reviewer #2: (No Response)

6. Review Comments to the Author

Reviewer #1: no additional questions/comments as previously raised ones were addressed, the manuscript has been thoroughly revised.

Reviewer #2: After re-consideration by the authors, the data reported in this manuscript deserves to be published in PLOS-ONE.

The authors conscientiously took into account the concerns and remarks of the reviewers, in particular concerning the description of the procedures and the origin with the characterization of the strains tested. The writing has been greatly improved. It remains 3 minor typing mistakes:

"C. auris" (l.139), "C. albicans" (l.141-142) and "Candida" (l. 145) must be in italic.

7. PLOS authors have the option to publish the peer review history of their article (what does this mean?). If published, this will include your full peer review and any attached files.

Reviewer #1: No

Reviewer #2: No

---

## [Editor Report · Acceptance letter]

1 May 2020

PONE-D-19-35054R1 

Abrogation of pathogenic attributes in drug resistant Candida auris strains by farnesol 

Dear Dr. Ahmad:

I am pleased to inform you that your manuscript has been deemed suitable for publication in PLOS ONE. Congratulations! Your manuscript is now with our production department. 

With kind regards,

on behalf of

Dr. Joy Sturtevant 

Academic Editor

PLOS ONE